# Extended Kalman Filter design for sensorless sliding mode predictive control of induction motors without weighting factor: An experimental investigation

**Mohamed Chebaani[1], Mohamed Metwally Mahmoud** [ID][2]*, **Ahmad F. Tazay[3], Mohamed I. Mosaad[4,5], Noura A. Nouraldin[6]**

**1** Department of Electrical Motorering, LGEB laboratory, Biskra University, Biskra, Algeria, **2** Electrical Engineering Department, Faculty of Energy Engineering, Aswan University, Aswan, Egypt, **3** Electrical Engineering Department, Colleague of Engineering, Al Baha University, Al Baha, KSA, **4** Electrical & Electronics Engineering Technology Department, Royal Commission Yanbu Colleges & Institutes, Yanbu Industrial City, Saudi Arabia, **5** Electrical Engineering Department, Faculty of Engineering, Damietta University, Damietta, Egypt, **6** Electrical Department, Faculty of Technology and Education, Suez University, Suez, Egypt

* metwally_m@aswu.edu.eg

**Data Availability Statement:** All relevant data are within the paper.

## Abstract

Due to their simplicity, cheapness, and ease of maintenance, induction motors (IMs) are the most widely used motors in the industry. However, if they are not properly controlled, the load torque and motor speed will fluctuate in an unsatisfactory fashion. To effectively control the load torque and speed of these IMs, it is necessary to use specialized drives. The entire system (IMs + Drives) will experience uncertainty, nonlinearities, and disruptions, which calls for an outstanding performance control structure. The sensorless sliding mode predictive torque control (SSM-PTC) for both AC-DC converter and DC-AC inverter, which are utilized for feeding the IM, is investigated in this work. The AC-DC converter is controlled using the SSM-PTC method in order to follow the DC-link reference voltage throughout any changes in the operating point of the IM. While the DC-AC inverter is controlled using a sensorless predictive power control (SPPC). Within a unity power factor, this SPPC regulates the reactive power flow between the motor and the supply to account for the undesirable harmonic components of the grid current. In addition, an experimental performance improvement of SSM-PTC of IM supplied by a 5-leg AC-DC-AC power converter using extended Kalman filter (EKF) without weighting factor (WF) is also studied in this work. Design and implantation of the suggested control systems are performed using a dSPACE 1104 card. The experimental results of the proposed converter control demonstrate that the suggested approach effectively regulated the DC link, reducing load torque and speed fluctuations. In the context of inverter control, a prompt active power response yields a motor current waveform that resembles a sinusoidal pattern, exhibiting minimal levels of harmonic distortion.

**Funding:** The author(s) received no specific funding for this work

**Abbreviations:** $\vec{v}_s$, Stator voltage vector; $\vec{i}_s$ $\vec{i}_r$, Stator, Rotor current vector; $\vec{\psi}_s$ $\vec{\psi}_r$, Stator, Rotor flux Stator voltage vector; $T_e$, Electromagnetic torque; $T_l$, Load torque; $\omega_m$, Rotor angular speed; $\omega_e = p\omega_m$, $\omega_e$ Rotor angular frequency; $V_{dc}$, DC-link voltage; $T_e^*$, torque reference; PI, Proportional-Integrator controller; $R_s,R_r$, Stator, rotor resistance; $L_s,L_r$, Stator, Rotor inductance; $L_m$, Mutual inductance; J, Moment of inertia; p, Number of pole pairs; $T_s$, Sampling time; $S_x(k+1)$, Switching state for the next time instant $(k+1)$; $\{v_0 \ldots v_7\}$, Possible voltage vectors; $k_r = \frac{L_m}{L_r}$, Rotor coupling factor; $R_\sigma = R_s + k_r^2 R_r$, Equivalent resistance referred to stator; $\tau_\sigma = \frac{L_\sigma}{R_\sigma}$, Transient time stator constant; $L_\sigma = \sigma L_s$, Leakage inductance; $\tau_r = \frac{L_r}{R_r}$, Rotor time constant; $\sigma = 1 - \frac{L_m^2}{L_s L_r}$, Total leakage factor; $\lambda_p$, Stator flux weighting factor; $\lambda_n$, Switching frequency weighting factor; Te*(k+1), Predicted reference torque; Tep(k+1), Predicted torque; $\vec{\psi}_s^*$, Reference stator flux; $\vec{\psi}_s^p$, Predicted stator flux; $i_{dc1},i_{dc2}$, The upper and lower DC link currents.

## 1. Introduction

Induction motors (IMs) are gaining increasing attention due to their extensive range of speed variations, mechanical durability, ease of maintenance, and notably their superior affordability in comparison to other electrical machines. Around 56% of the electrical energy consumption can be attributed to electric motors. Furthermore, an estimated 96% of this energy is consumed by IMs. Consequently, it can be inferred that almost 53% of the entire electrical energy consumption is accounted for by this type of motor. Also, approximately 70% of machine is used as induction motors for small power (<10 kW) [1–3]. Accordingly, the tendency is to concentrate the energy optimization studies on low-power IMs since the losses in these motors are of a value significant with their nominal power. These statistics clearly show the strong domination of the IMs for electrical machines in contemporary times. Additionally, according to statistical forecasters, it has been projected that IMs will continue to be the predominant machine in use for a minimum of twenty years [4–6].

In the past, most of the industrial regulations have often used analog regulator type (P.I. or P.I.D.) with remarkable efficiency and a reported price/performance that presents a formidable challenge for competitors. However, this type of regulator does not adequately address all requirements, and its performance is compromised in specific fields of applications [7–9]. The challenges associated with these controllers include several aspects, such as adjusting their parameters via the tuning mechanism, the presence of non-linearities within the system, and the potential for instability. When the performance is strained by the user, including strong mitigation of disturbance, following error to zero in pursuit, and response in minimum time, it leads to operating under constraints that affect either the command or the internal variables of the process. In addition to the limitations of regulators, conventional analog cited previously, the evolution of the digital electronic that allowed computers very fast has promoted the emergence of predictive control on the basis of a numerical model since, on the technical plan, accessibility is more comfortable for digital computers likely to realize of algorithmic treatments, integrating calculation and logical that regulators are purely analog [10–12].

Power electronic elements have made significant advancements recently and have the capacity to convert electrical energy. The studies conducted in this area consider several factors involving the topologies, constructions, and functionality of the power switches as well as control systems [13–15]. Among the significant occurrences causing the efficiency of the power to degrade, namely, the distortion of the voltage wave (VW), is the existence of harmonics in the electrical system, also known as harmonic contamination. The distortion is caused by the accumulation of sinusoidal waves (SW) with frequencies (Fs) different from the basic VW on top of its fundamentals. At non-multiple Fs of the F's fundamental, interharmonics are seen. This problem could harm electrical devices attached to the circuits and is frequently the result of improper electrical power utilization. The overheating of the neutral conductors of transformers and the degradation of capacitors result in harmonic contaminants, along with the premature activation of electrical safeguards and resonance occurrences with system parts [16, 17].

Utilizing diode rectifiers results in a high degree of harmonics produced in the system, which distorts the VW and worsens the grid-side power factor (PF). Active-front-end rectifier (AFER) toll, which can impose a sinusoidal source current for any load, control the PF, active power (P), and reactive power (Q), as well as ensure working recurrence are increasingly being used as substitutes for traditional rectifiers to prevent these imbalances [18]. A variety of drive systems employed this concept [19]. Through harmonic and Q compensation, the structure under study enables the use of a P's filter [20]. Achieving a regulated DC voltage ($V_{DC}$) is one of the AFER's primary goals. The capacitor charging\discharging method regulation, the $V_{DC}$ control loop's job is to keep it at a wanted value. In general, switching losses, inductance losses,

and variations in the load are the sources of $V_{DC}$ fluctuation. In order to regulate the transfer of P between the grid and the DC bus, the amplitude of the I's references was adjusted. For this reason, the energy saved in the capacitor on either the converter side or the load side is impacted base on interruptions [21, 22].

The scientific literature has suggested several different methods for controlling AFER. The similar goals of a high PF and a quasi-sinusoidal currents waveform are the focus of all these techniques. These approaches are divided into different categories based on the characteristics of the regulatory parameters (P/I). The voltage-oriented control (VOC), which employs current loops, is to align the voltage and current vectors in the same direction [23]. The direct power control (DPC) was created using the direct torque control (DTC) of IM. Utilizing two inner loops, DPC regulates the P&Q rather than the torque\flux [24]. In [25], two categories of current control techniques (linear and non-linear) have been used with PWM inverters [26]. In [27], controlling the currents was performed with fuzzy logic (FL). The concept relies on calculating a control vector using an FL, and the currents are regulated in the (α-β) frame. In several research papers, the VOC with no V/I sensors was discussed. In [28], a system's V estimation and an estimator of the currents taken from the DC bus were developed. Ref. [29], suggested a method for controlling current that relies on the idea of virtual flux and employed an established switching table.

Improving the performance of the control process is an ongoing problem prompting the development of more complex control strategies. Predictive control (PC) was one of the emerging techniques these last decades, not only in the academic plan but also in the industrial plan. It is the most used after the regulator's conventional analog (PI, PID). Although the PC was born of a real need in the industrial world and has seen the day within the oil and petrochemical industry, it is quickly propagated by touching of other industrial sectors after the great success in the oil industry proven seriously by the economic constraints caused by the oil crisis [30]. The PC philosophy is, therefore, knowing the output of the process to determine the command that allows him to the set point according to a predefined trajectory (path of reference) on the output of the process. Hence, it is imperative to determine the forthcoming commands for initiating the process entry to accomplish the desired consolidation. However, it should be noted that the initial command is executed only, as subsequent commands are disregarded due to the following sampling period. This periodicity causes the sequences to be staggered, resulting in a fresh output being measured. Consequently, the sequence of operations is reiterated at each sampling interval in conjunction with the horizon's orientation. In reality, the model of the process said internal model (implanted in the digital computer) does allow you to predict the evolution of its output since the model adopted is imperfect because of errors in the identification of disturbance not taken into account and the simplifications made allowing a use real-time. The result is that the output of the process is different from that of the model [31].

The first study of the PC approach of IMs was investigated in [32]. A PC of torque (PCT) technique has been presented to control the IM flux/ torque effectively while reducing the immediate reactive power at the input side [33]. The PC model (MPC) was suggested as an efficient method in high control performance of the power converters, for example, speed IM drives, matrix-converter, AC-DC-AC converter, and other configurations [34].

Over the past few years, commands without mechanical sensors (MSs) have received great attention. The basis for the training without MS is to estimate the IM's speed and position by the quantities of stator terminals measured. The estimation techniques often used for IMs can be classified as follows: an estimate of the flux by the voltage model; the estimators based on the model reference adaptive system (MRAS), extended Kalman filter (EKF), detection of distinctiveness based on the injection of high signal F; sliding mode observer. The EKF is an optimal estimator within the meaning of least squares to estimate the speed of IM using the V&I

measured [35]. The speed estimation based on the EKF requires powerful microprocessors to perform the complicated calculation. The model of the machine used is designated in a stationary repository and depends heavily on the parameters of the IM. In addition, the position of the initial rotor is not available. Because of these facts, the estimate of the speed built on the technique of the EKF is not really favored by many researchers.

The 5-legged AC-DC-AC converter architecture is provided in this study, along with an SSM-PTC-based control strategy for efficient IM control utilizing EKF with no WF. However, the system performance depends on the choice of WF in the cost function (CF). This study aims to address the issue of WF by proposing a solution that involves replacing the single CF with two distinct CFs, using a ranking technique. The current research has extensively examined the dynamic performance in terms of torque and flux ripples, the average F's switching, and reducing the incoming Q. Additionally, the suggested SSM-PTC's average switching F is decreased and is shown to only change in a very narrow F spectrum. MATLAB/Simulink and Controlling Desk programmers are used to build the suggested control technique on a dSPACE DS1104 R&D controller board. The proposed control mechanism efficiency is confirmed through experimentation under several operating conditions.

The structure of this work is as follows: After the introduction part, Section 2 presents the investigates system description and modeling. Section 3 presents the suggested control strategies (PTC, and power PC) and applications for the studied system. Furthermore, EKF for speed calculation is introduced in section 4. For the purpose of analyzing the performances of the prediction algorithms under consideration, experimental findings are provided in sections 5 and 6. Finally, section 7 concludes the main findings and obtained results from this study.

## 2. Investigates system modeling and explanation

Fig 1 shows the configuration of a 5-leg AC-DC-AC power converter. The converter can be considered as a 3-phase active front-end rectifier and a 4 switch three-phase inverter. The AFER operates as a chopper with AC voltage at the input, and $V_{DC}$ at the output. The 4-switch inverter receives $V_{DC}$ and converts it to AC voltage for the motor drive. The detailed $\alpha\beta$ model [20] will be explained in the next section.

### 2.1. Modeling of AFER

The 3-phase feed voltages ($v_g$) are linked to the rectifier's entirely regulated bridge of power transistors through the filter's inductances ($L_g$) and resistances ($R_g$). The vector formula

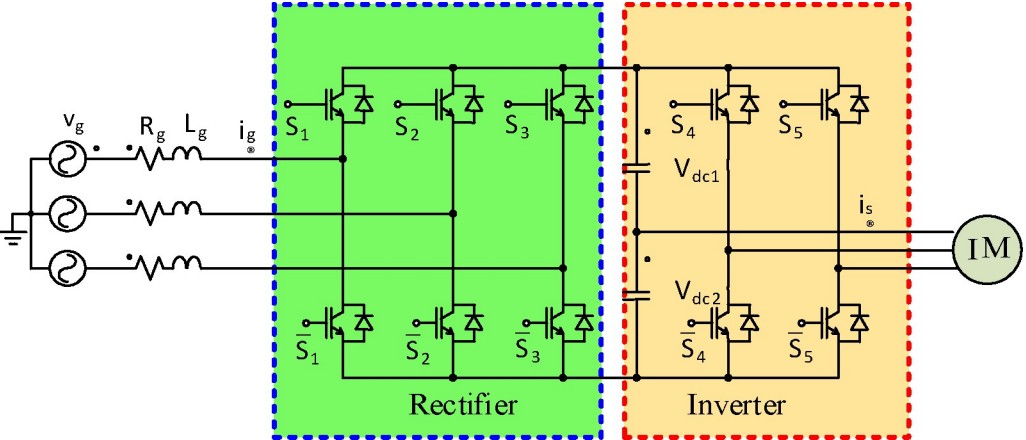

**Fig 1. Power circuit of the investigated configuration.**

indicates the incoming current in the $\alpha\beta$ frame [20, 36].

$$L_g \frac{d\vec{i}_g}{dt} = \vec{v}_g - \vec{v}_{afe} - R_g \vec{i}_g \tag{1}$$

where $\vec{i}_g = i_{g\alpha} + j \cdot i_{g\beta}$, $\vec{v}_g = v_{g\alpha} + j \cdot v_{g\beta}$, $\vec{v}_{afe} = v_{afe\alpha} + j \cdot v_{afe\beta}$ represents the incoming I's vector, V's supply line, and V's produced from the converter, respectively.

The formula links the phase Is and the incoming I's vector is presented as:

$$\vec{i}_g = \frac{2}{3}\left(i_{ga} + ai_{gb} + a^2 i_{gc}\right), \text{ and } a = e^{j2\pi/3} \tag{2}$$

Voltages $v_g$ and $v_{afe}$ are defined in a similar way

$$\vec{v}_g = \frac{2}{3}\left(v_{ga} + av_{gb} + a^2 v_{gc}\right) \tag{3}$$

The voltage $v_{afe}$ is determined by the converter switching state (SS) and the $V_{DC}$, and is expressed as

$$v_{afe} = V_{dc}S_{afe} \tag{4}$$

where $S_{afe}$ is the vector's SS of the rectifier defined as

$$S_{afe} = \frac{2}{3}\left(S_1 + aS_2 + a^2 S_3\right) \tag{5}$$

where $S_i, i = 1,2,3$ are the SSs of the rectifier 3 legs, $S_i = 1$ means ON switch and $\bar{S}_i = 1$ signifies OFF switch.

## 2.2. Modeling of a 4-switch 3- Ø inverter

The four studied 4-switch 3-Ø inverter contains 2-legs and the $V_{DC}$ is divided into 2 sources as seen in Fig 1. The switches in the systems are well-thought-out ideal. Hence, SSs $S_{inv}$ is written as [32, 37]:

$$S_{inv} = \frac{2}{3}\left(S_4 + aS_5 + 1\right) \tag{6}$$

The V's vector $v_s$ is associated with the SS $S_{inv}$ as seen:

$$v_s = V_{dc}S_{inv} \tag{7}$$

Table 1 lists the 4 potentials $v_s$ along their respective equivalent SS.

The fact that higher switching Fs can need a more complex converter model should be considered. For instance, it might cover diode forward V loss, transistor saturation V, and dead time modeling. Nevertheless, simplification is emphasized in this study, so a straightforward model of the inverter is utilized.

**Table 1. The $v_s$ and SS of a 4-switch 3- Ø inverter.**

| $v_n$ | $S = [S_4, S_5]$ | $v_s = v_\alpha + jv_\beta$ |
|---|---|---|
| $v_0$ | [0,0] | $2/3V_{dc2}$ |
| $v_1$ | [1,0] | $(V_{dc2} - V_{dc1})/3 - j\sqrt{3}(V_{dc1} + V_{dc2})/3$ |
| $v_2$ | [1,1] | $(V_{dc2} - V_{dc1})/3 + j\sqrt{3}(V_{dc1} + V_{dc2})/3$ |
| $v_3$ | [0,1] | $-2/3V_{dc2}$ |

## 2.3. Modeling of IM

The next Eqs., define the mathematical representation of an IM and is fully defined in [38, 39]:

$$\vec{v_s} = R_s \vec{i_s} + \frac{d\vec{\psi_s}}{dt} \tag{8}$$

$$0 = R_r \vec{i_r} + \frac{d\vec{\psi_r}}{dt} - j\omega_e \vec{\psi_r} \tag{9}$$

$$\vec{\psi_s} = L_s \vec{i_s} + L_m \vec{i_r} \tag{10}$$

$$\vec{\psi_r} = L_m \vec{i_s} + L_r \vec{i_r} \tag{11}$$

$$T_e = \frac{p}{2} \Im m \left\{ \vec{\psi_s^*} \cdot \vec{i_s} \right\} \tag{12}$$

$$J \frac{d\omega}{dt} = T_e - T_l - f\omega \tag{13}$$

## 3. Suggested control technique

As mentioned before, the proportional-integral (PI) controller is known for its simplicity and directness; nonetheless, it does possess significant limitations. These limitations may be mitigated by implementing certain adaptations. A basic implementation, low processing resource need, and reliance on maximum effort control characterize the SMC technique for adapting the PI controller [40]. The SMC will modify the PI controller to ensure its suitability and effectiveness in the presence of variations and uncertainties in system parameters. The SMC demonstrates prompt responsiveness to changes in system inputs, robustness against parameter variations, and the existence of non-linear system features [40].

In Fig 2(A), the suggested control strategy is depicted. The rectifier and inverter are under the PC system's supervision. To regulate the $V_{DC}$ and IM speed, respectively, PI and SM

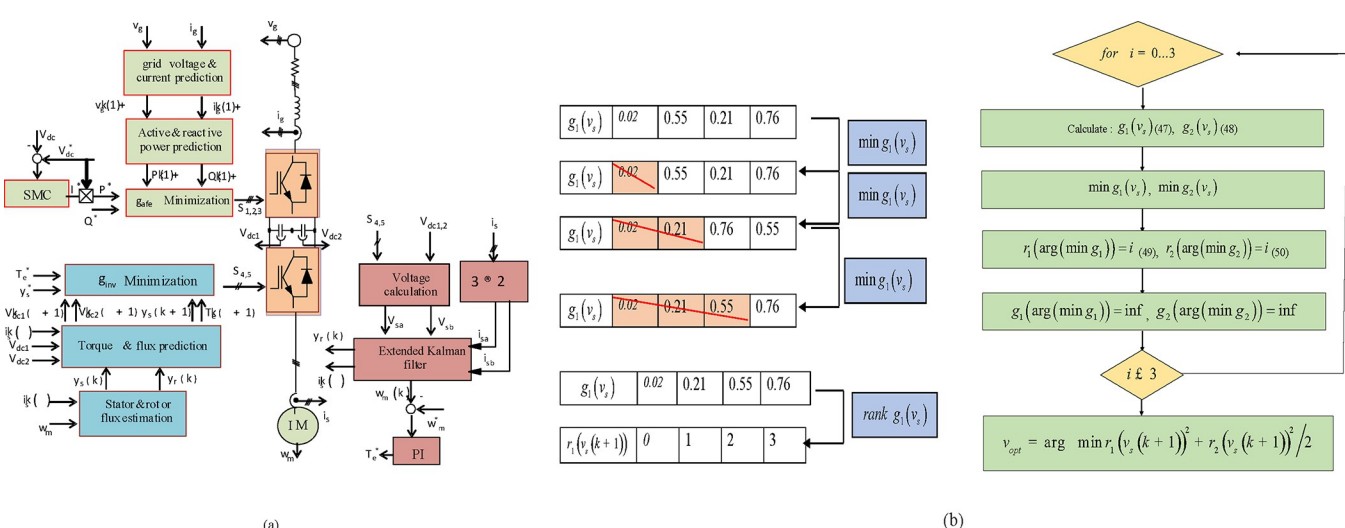

**Fig 2.** (a) Suggest control scheme of the addressed system, (b) The ranking strategy steps.

controllers are designed and implemented. Additionally, the PC can be used to control the $V_{DC}$ [41]. However, it can be inferred from the results that using a PC to regulate the $V_{DC}$ does not offer any advantages over a traditional SMC [40]. Speed PC is now being developed for the IM speed regulator.

### 3.1. Rectifier PPC

**3.1.1. Discrete-time prediction model (DTPM).**   A DTPM is necessary for applying PPC. The first-order Euler forward approximation approach results in a small sample period, which can be expressed as [42].

$$\frac{d\,\overrightarrow{i}_{g}^{p}}{dt} \approx \frac{\overrightarrow{i}_{g}^{p}(k+1) - \overrightarrow{i}_{g}^{p}(k)}{T_s} \tag{14}$$

where $k$ is the sampling instantaneous

The next anticipated current DT is produced by applying (14) to (1) which can be expressed as.

$$\overrightarrow{i}_{g}^{p}(k+1) = \left(1 - \frac{R_g T_s}{L_g}\right)\overrightarrow{i}_{g}(k) + \frac{T_s}{L_g} \times \left[\overrightarrow{v}_{g}(k) - \overrightarrow{v}_{afe}(k)\right] \tag{15}$$

The immediate incoming P&Q predictions are produced by considering the V&I vectors in orthogonal coordinates.

$$P(k+1) = v_{s\alpha}(k+1)i_{s\alpha}(k+1) + v_{s\beta}(k+1)i_{s\beta}(k+1) \tag{16}$$

$$Q(k+1) = v_{s\beta}(k+1)i_{s\alpha}(k+1) + v_{s\alpha}(k+1)i_{s\beta}(k+1) \tag{17}$$

**3.1.2. Reimbursement for the control delay.**   A significant number of computations are needed to apply PC in a practical system, which causes an actuation time delay that needs to be made up for. The compensation is carried through using two-step predictions. It has become necessary to forecast how the I will behave at the exact time ($k$+1), given that the chosen V would be given ($k$+2). The formula $i_g$($k$+2) is provided by time-shifting (15) forward a single step.

$$\overrightarrow{i}_{g}^{p}(k+2) = \left(1 - \frac{R_g T_s}{L_g}\right)\overrightarrow{i}_{g}(k+1) + \frac{T_s}{L_g} \times \left[\overrightarrow{v}_{g}(k+1) - \overrightarrow{v}_{afe}(k+1)\right] \tag{18}$$

where $i_g$($k$+2) is determined taking into account the converter V $v_{afe}(k)$ chosen during the prior sampling moment and utilizing the I&V measurements. The V $v_{afe}(k+1)$ that must be used is the V's converter.

**3.1.3. CF minimization.**   The CF assesses the mistake in the input Ps since the incoming P&Q for the rectifier is regulated. The CF summed up the rectifier's intended behavior $g_{afe}$, which includes bringing the P&Q up to typical values ($P^*$&$Q^*$) [43].

$$g_{afe} = |Q^* - Q(k+2)| + |P^* - P(k+2)| \tag{19}$$

In order to achieve a PF operation of unity, the $Q^*$ is often set to zero. But in some cases, it's possible to be non-zero. The CF for median SF minimization includes a switching transition

term, which is provided by:

$$H_{sw} = \sum_{x=\{a,b,c\}} |s_x(k+1)|_i - s_x(k) \tag{20}$$

The CF $g_{afe}$ must contain $I_{gm}$ to protect from over I. So, $I_{gm}$ is given as

$$I_{gm} = \begin{cases} \infty & if \quad |\vec{i}_g^p(k+1)| > I_{gmax} \\ 0, & otherwise \end{cases} \tag{21}$$

Consequently, the PC's whole CF is

$$g_{afe} = |Q^* - Q(k+2)| + |P^* - P(k+2)| + \xi_n H_{sw} + I_{gm} \tag{22}$$

where $\xi_n$ is the WF of $H_{sw}$. The $I_{gm}$ does not need any WF.

**3.1.4. Control of $V_{DC}$ with SMC.** An SMC is used for $V_{DC}$ management in order to achieve appropriate dynamic response and SSP of the rectifier. The SMC's outputs are in line with the amount of P required to update the error in the $V_{DC}$. The process for the SMC command is as outlined below:

$$C_{eq} \frac{dV_{dc}}{dt} = i_{dc} - i_{out} \tag{23}$$

The $V_{DC}$ mistake can be written as

$$e_v = V_{dc}^* - V_{dc} \tag{24}$$

The formula for the Lyapunov function is

$$V = \frac{1}{2} C_{eq} e_v^2 \tag{25}$$

If $V_{dc}^*$ is constant, then

$$\dot{V} = e_v(i_{out} - i_{dc}) \tag{26}$$

If it is assumed that $|i_{out}| < I_{max}$, we can take the order

$$i_{dc} = I_{max} \text{sign}(e_v) \tag{27}$$

Then we have

$$\begin{aligned} \dot{V} &= e_v(i_{out} - I_{max}\text{sign}(e_v)) \\ &= |e_v|(i_{out}\text{sign}(e_v) - I_{max}) \end{aligned} \tag{28}$$

As $|i_{out}| < I_{max}$ then from $(i_{out}\text{sign}(e_v) - I_{max}) < 0$ where $\dot{V} < 0$ and $e_v$ asymptotically approaches zero.

To execute, we make use of:

$$i_{dc} = I_{max} \frac{e_v}{|e_v| + \varepsilon} \tag{29}$$

With $\varepsilon$ minor positive constant that congregates to a variety $|e_v| < \varepsilon$).

## 3.2. SPTC of IM

The 3-phase that make up the SPTC of IM are flux estimate, flux and torque prediction, and CF reduction.

**3.2.1. Flux and torque calculations.** The effectiveness of DTC-based control techniques largely depends on the precise estimation of stator flux, which is accomplished by using stator Vs&Is. There are two distinct groups of stator flux estimation methods, as specified by (8) and (9), that are based on the V&I model correspondingly. Comparatively, fewer variables are needed for the V's model calculator than for the I's model calculator. The DC drift of I sensors prevents the perfect integrator in (8) from operating correctly in an actual setting. I sensors and signal conditioning circuits will always experience DC drifts when measuring stator Is. The machine drive system becomes unstable as a result of the accumulation of errors brought on by DC drift during the integration phase. Instead of the ideal integrator, the low pass filter (LPF) is the most frequently used approach. The LPF can always carry out the integration task under typical operational circumstances. When the signal is DC, the gain for LPF compensating and the filter time constants become unlimited, making it impossible to conduct the integration. The magnitude and phase angle faults introduced by LPF render the controller more complicated due to the extra measures required to correct those defects. In this study, the flux's rotor and stator are approximated using the IM I model, nevertheless, the speed and stator Is are calculated rather than measured [44].

$$\frac{d\vec{\hat{\psi}}_r}{dt} = R_r \frac{L_m}{L_r} \vec{i}_s - \left(\frac{R_r}{L_r} - j\omega_e\right) \vec{\hat{\psi}}_r \tag{30}$$

$$\vec{\hat{\psi}}_s = \frac{L_m}{L_r} \vec{\hat{\psi}}_r + \sigma L_s \vec{i}_s \tag{31}$$

The torque estimation is given by

$$T_e = 1.5 p \Im m\{\vec{\psi}_s^* . \vec{i}_s\} \tag{32}$$

**3.2.2. DTPM for IM.** The discrete forms of (30) and (31) are generated by the backward-Euler approach.

The discrete expressions for Eqs (30) and (31) are derived using the backward-Euler method and can be represented as:

$$\vec{\hat{\psi}}_r(k) = \vec{\hat{\psi}}_r(k-1) + T_s\left[R_r \frac{L_m}{L_r} \vec{i}_s(k) - \left(\frac{R_r}{L_r} - j\omega_e(k)\right)\vec{\hat{\psi}}_r(k-1)\right] \tag{33}$$

$$\vec{\hat{\psi}}_s(k) = \frac{L_m}{L_r} \vec{\hat{\psi}}_r(k) + \sigma L_s \vec{i}_s(k) \tag{34}$$

Therefore, the determined torque is expressed as follows:

$$\hat{T}_e(k) = 1.5 p \Im m\{\vec{\hat{\psi}}_s^*(k) . \vec{i}_s(k)\} \tag{35}$$

**3.2.3. Torque predictions and stator flux.** At the sampling phase (k+1), the stator flux and torque must be anticipated. Stator V model typically uses the stator flux anticipating, which can be provided in discrete intervals of time as:

$$\vec{\psi}_s^p(k+1) = \vec{\hat{\psi}}_s(k) + T_s \vec{v}_s(k) - T_s R_s \vec{i}_s(k) \tag{36}$$

Additionally projected are the stator I and torque. Consequently, the stator I and torque

predictions are expressed as:

$$\overrightarrow{i}_s^p(k+1) = \left(1 + \frac{T_s}{\tau_\sigma}\right)\overrightarrow{i}_s(k) + \frac{T_s}{(\tau_\sigma + T_s)} \times \left\{\frac{1}{R_\sigma}\left[\left(\frac{k_r}{\tau_r} - k_r j\omega_e(k)\right)\overrightarrow{\hat{\psi}}_r(k) + \overrightarrow{v}_s(k)\right]\right\} \quad (37)$$

$$T_e^p(k+1) = 1.5p\Im m\{\overrightarrow{\hat{\psi}}_s^p(k+1)^* . \overrightarrow{i}_s^p(k+1)\} \quad (38)$$

**3.2.4. CF minimization.** The anticipated variables were valued according to a preset CF, involving the torque error $(T_e^* - T_e^p)$ and flux errors' $(\overrightarrow{\psi}_s^* - \overrightarrow{\psi}_s^p)$ precise values.

$$g = |T_e^*(k+1) - T_e^p(k+1)| + \lambda_p||\psi_s^*| - |\psi_s^p(k+1)|| \quad (39)$$

The CF for reducing average SF contains a switching changeover term that can be expressed as:

$$n_{sw} = \sum_{x=\{a,b,c\}} |S_x(k+1)_i| - S_x(k) \quad (40)$$

The CF g must contain another term $I_m$ to avoid over-current. Consequently, the $I_m$ is definite as $I_m = \begin{cases} \infty & if \quad |\overrightarrow{i}_s^p(k+1)| > I_{\max} \\ 0 & therwise \end{cases}$

Consequently, the controller's whole CF's g is

$$g = |T_e^*(k+1) - T_e^p(k+1)| + \lambda_p||\overrightarrow{\psi}_s^*| - |\overrightarrow{\psi}_s^p(k+1)|| + \lambda_n n_{sw} + I_m \quad (41)$$

The most efficient V vector ($v_{opt}$) is chosen and applied to the IM terminal by the inverter at the following sampling instant. This V vector produces the least amount of g. When a control method is implemented, a single time delay must be made up for [45]. It is carried out using a 2-step prediction. Thus, the ideal voltage vector is chosen by minimizing the next CF.

$$g = |T_e^*(k+2) - T_e^p(k+2)| + \lambda_p||\overrightarrow{\psi}_s^*| - |\overrightarrow{\psi}_s^p(k+2)|| + \lambda_n n_{sw} + I_m \quad (42)$$

**3.2.5. Offset compensation of $V_{DC}$.** The 2-capacitor voltage may deviate in the opposite direction until the converter shuts down due to the phase 'a' I's incorrect beginning phase angle or an unbalanced current caused by speed variation. Therefore, adjustment of the $V_{DC}$ deviation is required. A function of the switching states is used to determine the DC-link's Is.

$$\begin{aligned} i_{dc1} &= i_b \cdot S_b + i_c \cdot S_c \\ i_{dc2} &= i_b \cdot (1 - S_b) + i_c \cdot (1 - S_c) \end{aligned} \quad (43)$$

The capacitor voltages are obtained:

$$\begin{aligned} C_1 \frac{dV_{dc1}}{dt} &= i_{dc1} - i_{out1} \\ C_2 \frac{dV_{dc2}}{dt} &= i_{dc2} - i_{out2} \end{aligned} \quad (44)$$

So, it is possible to determine the anticipated capacitor V by

$$V_{dc1}(k+1) = V_{dc1}(k) - (T_s/C_1)(i_{dc1}(k) - i_{out1}(k))$$
$$V_{dc2}(k+1) = V_{dc2}(k) - (T_s/C_2)(i_{dc2}(k) - i_{out2}(k))$$

(45)

where $C_1, C_2$ are the higher and lesser capacitance.

The $V_{dc1}(k+2)$ and $V_{dc2}(k+2)$ are gotten similarly. By inserting an additional term to the CF (42) we obtain the CF with the V offset mitigation.

$$g = |T_e^*(k+2) - T_e^p(k+2)| + \lambda_p||\overrightarrow{\psi}_s^*| - |\overrightarrow{\psi}_s^p(k+2)|| + \lambda_n n_{sw}$$
$$+ \lambda_{dc}\frac{|V_{dc1}(k+2) - V_{dc2}(k+2)|}{V_{dc}} + I_m$$

(46)

where $\lambda_{dc}$ is the WF of the $V_{DC}$'s capacitor offset compensation.

**3.2.6. WFs elimination.** The choice of $\lambda_p$ value is a challenging task that greatly affects the controller's effectiveness. The WF $\lambda_p$ must be removed from the equation to fix this issue. The proposed approach addresses the problem of selecting the V vector in classic PTC, which employs two distinct cost functions:.

$$g_1 = |T_e^*(k+2) - T_e^p(k+2)|$$

(47)

$$g_2 = ||\overrightarrow{\psi}_s^*| - |\overrightarrow{\psi}_s^p(k+2)||$$

(48)

where $g_1$ and $g_2$ are the mistakes of the torque and stator flux.

This approach was built to evaluate these components independently for the converter's eight V vectors. The steps of the strategy are as follows. Firstly, the error values obtained from the calculation of two CFs $g_1$ and $g_2$ are ranked, V vectors with lesser error receive a less favorable ranking, whereas those that have more error receive an improved ranking.

$$g_1(v_s(k+1)) \rightarrow r_1(v_s(k+1))$$

(49)

$$g_2(v_s(k+1)) \rightarrow r_2(v_s(k+1))$$

(50)

where $r_1(v_s(k+1))$ and $r_2(v_s(k+1))$ are the ranking values allied with $g_1$ and $g_2$.

According to Fig 2(B), the suggested solution ranking technique is explained in the order listed below. Due to its fundamental nature, the suggested approach does not need a substantial computational expense. Second, choose the V vector with the lowest ranking, resulting in an equivalent tracking of variables, torque, and flux. The optimization ranking is then displayed as

$$v_{opt} = \arg \quad \min\frac{r_1(v_s(k+1))^2 + r_2(v_s(k+1))^2}{2}$$

(51)

An illustrative example is provided in Table 2 to elaborate the proposed algorithm. The torque error is obtained by the voltage $v_0$ is $g_1(v_0) = 0.02$, the place value given is $r_1(v_0) = 0$ for the reason that its error value is smaller than that of the torque error obtained by $v_4$ is $g_1(v_3) = 0.76$, and then, the ranking assigned is $r_1(v_3) = 3$ as the error number is higher.

The flow errors should be evaluated using the same technique and assigned a corresponding ranking $r_2$. Certain voltage vectors might rank the same way, for example. To handle this problem, we choose the optimization presented in (51).

The lower value of the rankings $r_1 = 1$ and $r_2 = 0$ is provided by the V vector $v_2$. The V vector that will be used throughout the subsequent sample time is $v_2$.

**Table 2. Example of voltage vector selection.**

| $v_s$ | $g_1(v_s)$ | $g_2(v_s)$ | $r_1(v_s)$ | $r_2(v_s)$ | $r_1(v_s(k+1)) + r_2(v_s(k+1))/2$ | $r_1(v_s(k+1))^2 + r_2(v_s(k+1))^2/2$ |
|-------|-----------|-----------|-----------|-----------|-----------------------------------|----------------------------------------|
| $v_0$ | 0.02 | 0.72 | 0 | 3 | 1.5 | 4.5 |
| $v_1$ | 0.55 | 0.12 | 2 | 1 | 1.5 | 2.5 |
| $v_2$ | 0.21 | 0.06 | 1 | 0 | 0.5 | 0.5 |
| $v_3$ | 0.76 | 0.14 | 3 | 2 | 2.5 | 6.5 |

## 4. EKF for speed calculation

The Extended Kalman Filter proposed in this study demonstrates enhanced accuracy regarding steady state speed estimation inaccuracy. Additionally, it typically exhibits a lower standard deviation in these estimates, indicating a more stable and less oscillatory response. The Extended Kalman Filter is more effective at processing measurement noise than the Sliding Mode Observer. In the context of state estimation for the IMs, the authors posits that the EKF is a more judicious selection compared to the sliding Mode Observer. This assertion is contingent upon the feasibility of integrating the EKF while considering the computational capabilities of the dSPACE card.

The EKF is a state-optimum estimator that relies on unpredictable uncertainty associated with the system parameters; to generate the multi (input\output) structure, like the IM model, it employs device noise and statistics characteristics associated with measuring noise. In order to clarify the non-linear chaotic structures, the next Eq., is presented:

$$\begin{cases} \dot{x}(k) = f(x(k), u(k)) + w(k)(\text{System}) \\ y(k) = Cx(k) + v(k)(\text{Measurement}) \end{cases} \tag{52}$$

where $C$, $w(k)$, and $v(k)$ are the systems' output matrix, process noise, and measurement noise, respectively. The covariance matrices of noise's process and measurement are $Q$ and $R$.

The estimating process is described below:

- *State forecast\prediction*:

$$\dot{x}(k) = f(\dot{x}(k-1), u(k), x_0) \tag{53}$$

- *Covariance's calculation*:

$$P(k) = G(k)P(k-1)G(k)^T + Q \tag{54}$$

where:

$$G(k) = \frac{\delta f}{\delta x}(x(k-1), u(k-1), w(k-1)) \tag{55}$$

- *Kalman gain's calculation*:

$$K(k) = P(k)C(k)^T(C(k)P(k)C(k)^T + R)^{-1} \tag{56}$$

- *The state brings up-to-date*:

$$\dot{x}(k+1) = \dot{x}(k) + K(k)[y(k) - C(\dot{x}(k))] \tag{57}$$

- *Calculating estimating covariance*:

$$P(k) = (I - K(k))C(k)P^{-1}(k) \tag{58}$$

EKF is employed for state estimation and additional states like the speed that are incorporated into the state vector. The discontinuous and linear for Eq 52 is generated from the next Eq., taking into account the $w(k)$, and $v(k)$, and assuming the sampling time ($T_s$).

$$\begin{cases} x(k+1) = A_d x(k) + B_d u(k) + w(k) \\ y(k+1) = C_d x(k) + v(k) \end{cases} \tag{59}$$

where:

$$x = \begin{bmatrix} i_{s\alpha} & i_{s\beta} & \phi_{r\alpha} & \phi_{r\beta} & \omega_m \end{bmatrix}^T \tag{60}$$

$$A_d = \begin{bmatrix} -\dfrac{R_s}{L_\sigma} - \dfrac{L_m^2 R_r}{L_\sigma L_r^2} & 0 & \dfrac{L_m R_r}{L_\sigma L_r^2} & \dfrac{L_m \omega_e}{L_\sigma L_r} & 0 \\ 0 & -\dfrac{R_s}{L_\sigma} - \dfrac{L_m^2 R_r}{L_\sigma L_r^2} & -\dfrac{L_m \omega_e}{L_\sigma L_r} & \dfrac{L_m R_r}{L_\sigma L_r^2} & 0 \\ \dfrac{L_m R_r}{L_r} & 0 & -\dfrac{R_r}{L_r} & -\omega_e & 0 \\ 0 & \dfrac{L_m R_r}{L_r} & \omega_e & -\dfrac{R_r}{L_r} & 0 \\ 0 & 0 & 0 & 0 & 1 \end{bmatrix} \tag{61}$$

$$B_d = \begin{bmatrix} \dfrac{1}{L_\sigma} & 0 & 0 & 0 & 0 \\ 0 & \dfrac{1}{L_\sigma} & 0 & 0 & 0 \end{bmatrix} \quad C_d = \begin{bmatrix} 1 & 0 & 0 & 0 & 0 \\ 0 & 1 & 0 & 0 & 0 \end{bmatrix} \tag{62}$$

The EKF algorithm and using Eq 59, IM's speed is calculated by estimating the model state at various points in time. To achieve the target estimation performance, the covariance matrices Q and R are chosen through trial and error as shown as follows:

$$Q = diag\begin{bmatrix} 10^{-4} & 10^{-4} & 10^{-7} & 10^{-7} & 10^{-2} \end{bmatrix} \tag{63}$$

$$R = diag\begin{bmatrix} 10^{-2} & 10^{-2} \end{bmatrix} \tag{64}$$

## 5. Experimental design and implantation

The experimental apparatus is constructed in Biskra's (LGEB) electrical motoring lab, as depicted in Fig 3. It comprises 2 SIMEKRON converters, one of which serves as an inverter and the other as a rectifier, driving a 3-kW squirrel-cage IM through an AC-DC-AC converter. The motor shaft is connected to a permanent magnet synchronous machine (PMSM), which serves as the load. To change the load, a rheostat is connected in series with the PMSM's stator.

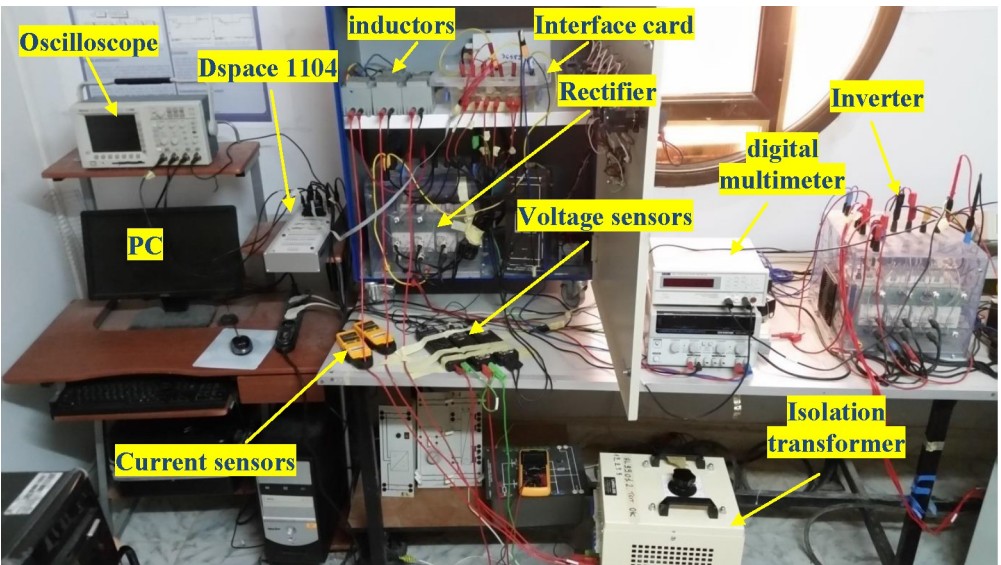

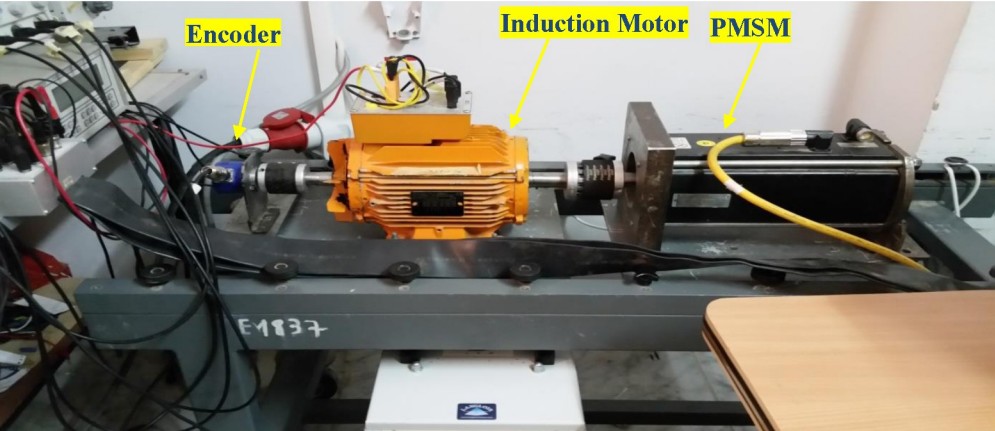

**Fig 3. System setup.**

A 1024-point cumulative encoder is used to determine the rotor's location in order to verify the speed's accuracy. A grid-connected autotransformer is used to supply the power to the motor. The Hall-effect I sensors detect 2 grid and load Is. The Hall-effect V sensors detect the source Vs and the V's capacitor. The dSPACE DS1104 R&D controller board is utilized to implement the control strategy in conjunction with the control desk and MATLAB Simulink programming suites. Table 3 presents the studied system parameters, which were derived using conventional experimental procedures. The information about the controller and the

**Table 3. Experimental setup parameters.**

| Rated power | $P_n = 3kW$ | Rated current | $I_n = 6.3A$ | DC-bus voltage | $V_{dc} = 450V$ |
|---|---|---|---|---|---|
| Stator resistance | $R_s = 2.3\Omega$ | Stator flux | $\psi_{snom} = 0.8Wb$ | Filter inductance | $L_g = 6.5mH$ |
| Rotor resistance | $R_r = 1.8\Omega$ | Rated Torque | $T_{nom} = 20Nm$ | Equivalent resistance | $R_g = 0.42\Omega$ |
| Stator inductance | $L_s = 0.261H$ | Number of the pole pairs | $Np = 2$ | Source voltage | $V_g = 140V$ |
| Rotor inductance | $L_r = 0.261H$ | Moment of inertia | $J = 0.03Kgm^2$ | DC-link capacitor | $C = 2040\mu F$ |
| Mutual inductance | $L_m = 0.258H$ | Rated speed | $\omega_m = 1415rpm$ | Source voltage frequency | $f_s = 50Hz$ |

**Table 4. Control parameters and load specifications.**

| Control parameters | Load specifications |
|---|---|
| $K_p = 0.4, k_i = 10$ <br> $\lambda_p = 100, \lambda_n = 0.05, \xi = 0.01$ <br> $I_{\max} = 15A, I_{gmax} = 20A$ | $P = 4\text{kW}, V = 400\text{V}$ <br> $I_a = 11A$ <br> $\omega = 3000\text{rpm}$ |

load parameters are given in Table 4. The $T_s$ is set at $130\mu s$ for calculation, forecasting, and actuating of the desired function. A time-based counter is used for the measurement of the execution time of the algorithms. To get the duration time of a certain step functions such as starting and reading the counter are employed. The built-in PowerPC time base has a high resolution of 4/bus clock (40 ns). Finally, the measured execution is observed in the Control Desk. The proposed algorithm requires additional calculations for the multi-objective ranking method and the dEKF observer. A technique for speed estimation based on EKFs was implemented to enhance the dynamic estimation performance and reduce the computational complexity. This approach involved utilizing a two-dimensional EKF working sequentially to address the inherent hysteresis observed in conventional EKFs. Therefore, the required maximum execution time is 115.6 μs. Most of the execution time is spent on the estimations (Torque and flux) and predictions: 27% and 62%, respectively. For this reason, executing the proposed control algorithm within 130 μs is possible.

# 6. Experimental results and discussions

Several experiments are conducted using the DS1104 to assess the efficacy of the proposed SSM-PTC approach.

## 6.1. Case 1: Speed reversal

The system's dynamic performance is checked in this scenario while performing a speed reverse maneuver. The dynamic behavior of the grid and motor side control schemes when the desired speed value varies within the range of 1000 to -1000 rpm, while subjected to a load torque of 5 Nm is depicted in Fig 4(A) and 4(B). To keep a PF of unity, the Q must be set to zero. Fig 4(A) displays the dynamic performance of the $V_{DC}$, P, Q, and phase. As can be observed, the SSM-PPC perfectly separates P&Q on the grid side during the transient. A low $V_{DC}$ volatility and a quicker P response are also characteristics of the SSM-PPC design.

The motor switches to generator mode throughout the speed reversal, and P is sent back to the grid. The IM enters motor mode and rapidly increases the P after 0.12s. The $V_{DC}$ changes by around 20 V during the load disturbance, demonstrating remarkable performance. The Q follows its benchmark. The PF is brought to unity. The I's grid as a result has a THD of 5.28% and is almost sinusoidal. Fig 4(B) shows the outcomes of the absolute and calculated speed, torque, flux, and I's phase. The outcomes demonstrate a flawless decoupling of flux and torque. High dynamic performance for the torque is provided by the PTC. With excellent performance and little overshoot, the estimator is able to monitor the speed. The stator flux significantly reduces ripples as it keeps the reference value (0.8 Wb). There are fewer ripples and greater dynamic torque. As a result, the I's stator has a THD of 4.75% and is almost sinusoidal.

## 6.2. Case 2: Steady-state at medium and low-speed operation

Fig 5(A) and 5(B) demonstrate, respectively, the steady-state performance (SSP) of the grid side (GS) and motor side (MS) for control methods at a standard speed of 1000 rpm and a load

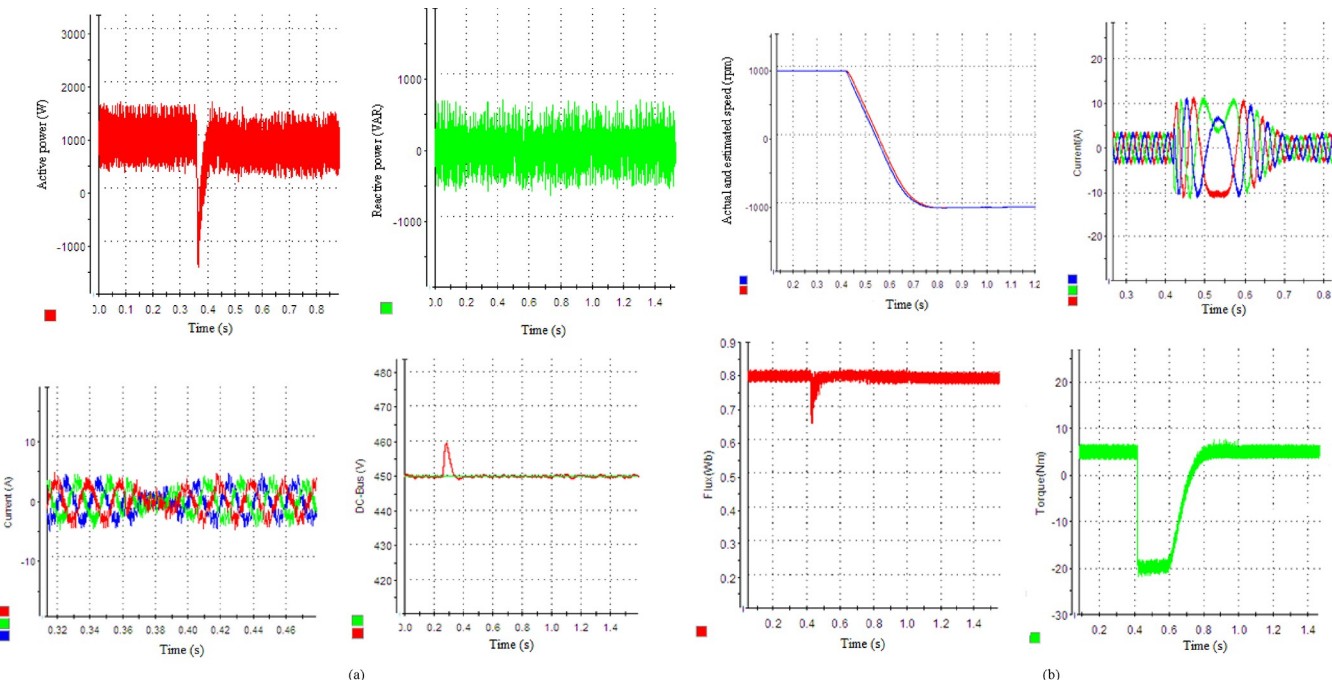

**Fig 4.** Systems' performance: (a) rectifier side through SSM-PPC technique (b) motor side through SPTC technique.

torque of 5 Nm. $V_{DC}$ reference is configured at 450V. It is obvious that the proposed approach can follow the prespecified P&Q. The PF is unity and the $V_{DC}$ smoothly maintains its desired value. Fig 5(B) shows the real and calculated speeds, torque, flux, and phase current profiles. Fast Fourier Transform (FFT) results for the I's phase spectra of the GS and MS are shown in Fig 5(C) and 5(D), accordingly. The grid and motor currents exhibit total harmonic distortion (THD) values of 4.65% and 4.01%, respectively. The F spectra reveal a tiny additional low-order harmonic in the grid current and a larger low-order harmonic in the motor phase currents.

The findings indicate that the SSM-PPC's median switching frequency (SF) $f_{swg}$ = 3.8 kHz and the $f_{swm}$ PTC's = 3.43 kHz. A switching transition term is inserted into the CF to lower the SSM-PPC and PTC's average SF. The F is then subjected to a WF. Selecting a suitable and optimal WF while maintaining a constant stator flux and Q errors presents a challenge. This issue arises because the amplified torque and power ripples occur when a greater WF is used.

The empirical validation of the alternative WF value, which effectively reduces the average SFs, is shown in Table 5.

To evaluate the effectiveness of the suggested controller for the IM at low-speed operating conditions, a rotational speed of 300 rpm is selected with a load torque of 5 Nm. The suggested scheme exhibits an excellent performance with an I's phase THD of 4.3% for SSM-PPC and 3.75% for SPTC, as depicted in Fig 6(A) and 6(B). Fig 6(C) and 6(D) show the F spectra of the I's grid and motor, respectively. The outcomes demonstrate that the average SF increases at low speeds. The SPTC is equal to 4.72 kHz, the SSM-PPC is equal to 3.8 kHz.

### 6.3. Case 3: Zero speed with variable load torque conditions

Another test case will be examined to assess the effectiveness of the proposed controller under a standstill condition with varying applied load torques. The motor is initially subjected to a

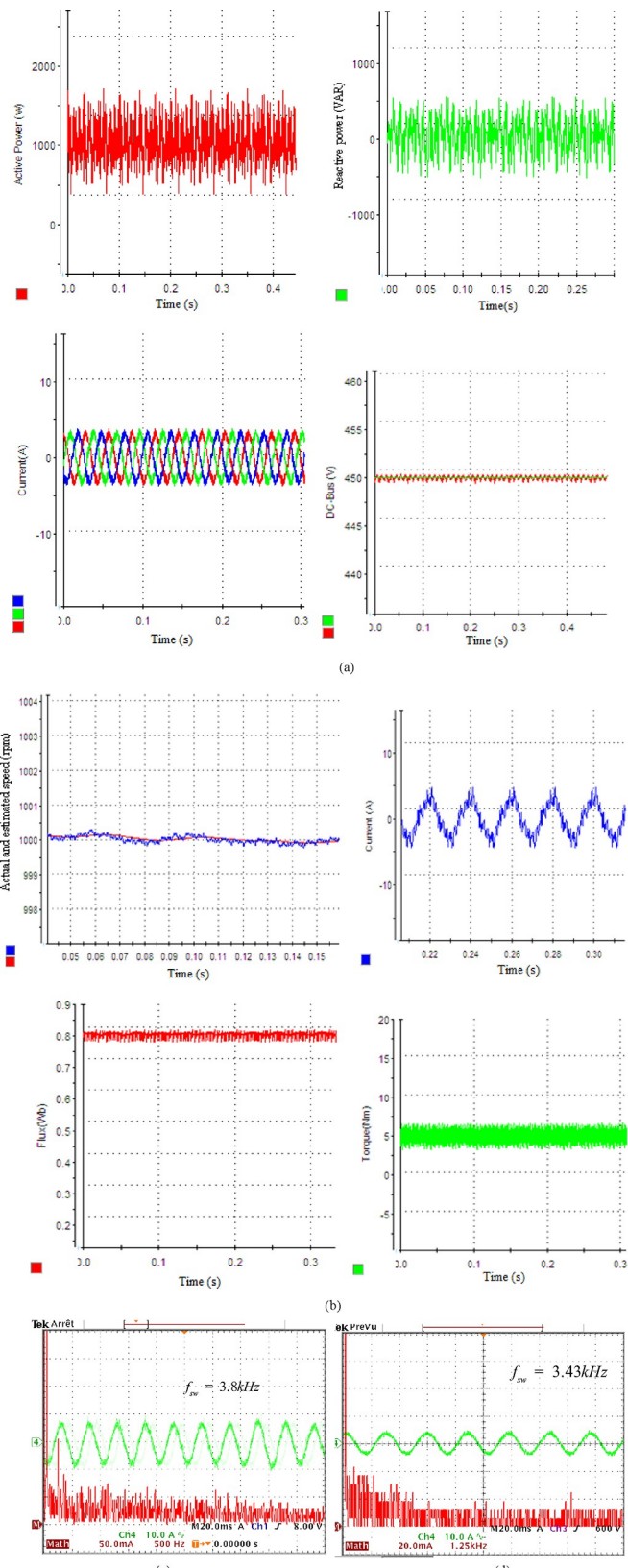

**Fig 5.** Experimental results display the SSP: (a) rectifier side with SSM-PPC scheme, (b) MS with SPTC scheme, (c) F spectra of I's grid, (d) F spectra of I's motor at 1000 rpm.

**Table 5. Results of SSP for different WF: $w_m = 1000rpm, T_l = 5Nm$.**

| WFs | $f_{swg}$ (Grid) kHz | $f_{swg}$ (Motor) kHz | P ripple (W) | T ripple (Nm) | THD (Grid) % | THD (Motor) % |
|---|---|---|---|---|---|---|
| $\lambda_n = \xi_n = 0$ | 4.25 | 3.85 | 370 | 2.5 | 4.3 | 3.75 |
| $\lambda_n = \xi_n = 0.01$ | 4.17 | 3.55 | 385 | 2.62 | 4.42 | 3.81 |
| $\lambda_n = 0.02, \xi_n = 0.015$ | 4 | 3.65 | 390 | 2.73 | 4.55 | 3.95 |
| $\lambda_n = 0.05, \xi_n = 0.03$ | 3.8 | 3.72 | 400 | 2.68 | 4.65 | 3.87 |
| $\lambda_n = 0.05, \xi_n = 0.01$ | 3.8 | 3.43 | 400 | 2.8 | 4.65 | 4.01 |

load torque of 3 N.m. Subsequently, the load change is simulated by different values of the load torque of 5, 8, and 10 N.m are simulated at 1, 5, and 8 s respectively. These simulations are conducted under zero-speed conditions. The responses to external load disturbance are illustrated in Fig 7(A) and 7(B), with the sudden change in the load torque The unity PF operation is successfully achieved during the dynamic process he unity PF operation is successfully achieved during the dynamic process by maintaining the Q null. The P tracks its new reference with acceptable stability and accuracy, exhibiting strong robustness against external load disturbance. The torque follows its further reference with satisfactory stability and accuracy, indicating strong robustness against motor load disturbance. The THD values of the grid and motor currents are 4.1% and 3.43%, respectively.

## 6.4. Case 4: System parameters variations ($R_s$, $R_r$, and $L_m$)

This study aims to explore the ability of the suggested control system to adapt to changes in system parameters. Changes in the values of Rs and Rr are employed to replicate the impact of temperature variations, while alterations in Lm are utilized to imitate changes in frequency. The effect of changes in these three parameters $R_s$, $R_r$, and $L_m$ is especially examined at 1000 rpm speed with 5 Nm. This choice is made due to the increased impact of parameter fluctuations on the accuracy of estimation inside these regions. The results are shown in Fig 8(A)–8(C). Upon evaluation of these experimental results, there is a slight change in the speed estimation of speed. Nevertheless, alterations in these parameters result in heightened load torque mistakes and faults in Flux calculation. In order to enhance the resilience of the observer in the face of parameter variations, it is possible to estimate these parameters by including them in the observer model. A comparison with previously published works is provided in Table 6 to show the effectiveness of the proposed strategy.

## 7. Conclusion

In this work, an SSM-PTC of IM supplied by a 5-leg AC-DC-AC power converter using EKF without WF is addressed for following the $V_{DC}$ for rectification control. The GS is controlled using an SPPC. The EKF was an effective method, and its effectiveness has been tested in various effective method, and its effectiveness has been tested in various settings with experimental data. The MS converter is coupled with the SPC of torque to lessen torque and flux ripples as well as the average SF. The control systems exhibit satisfactory performance with fast P response, low change in V, a unity power factor, and negligible torque and flux ripples. GS and MS of the booth experience a reduction in the average SF. According to the findings from the experiments, the utilized techniques provide a trustworthy estimation for a 3-Ø IM under various operating situations. Further study can be performed by presenting an innovative technique to estimate rotational speed, rotor, and stator resistances under uncertainty and fault.

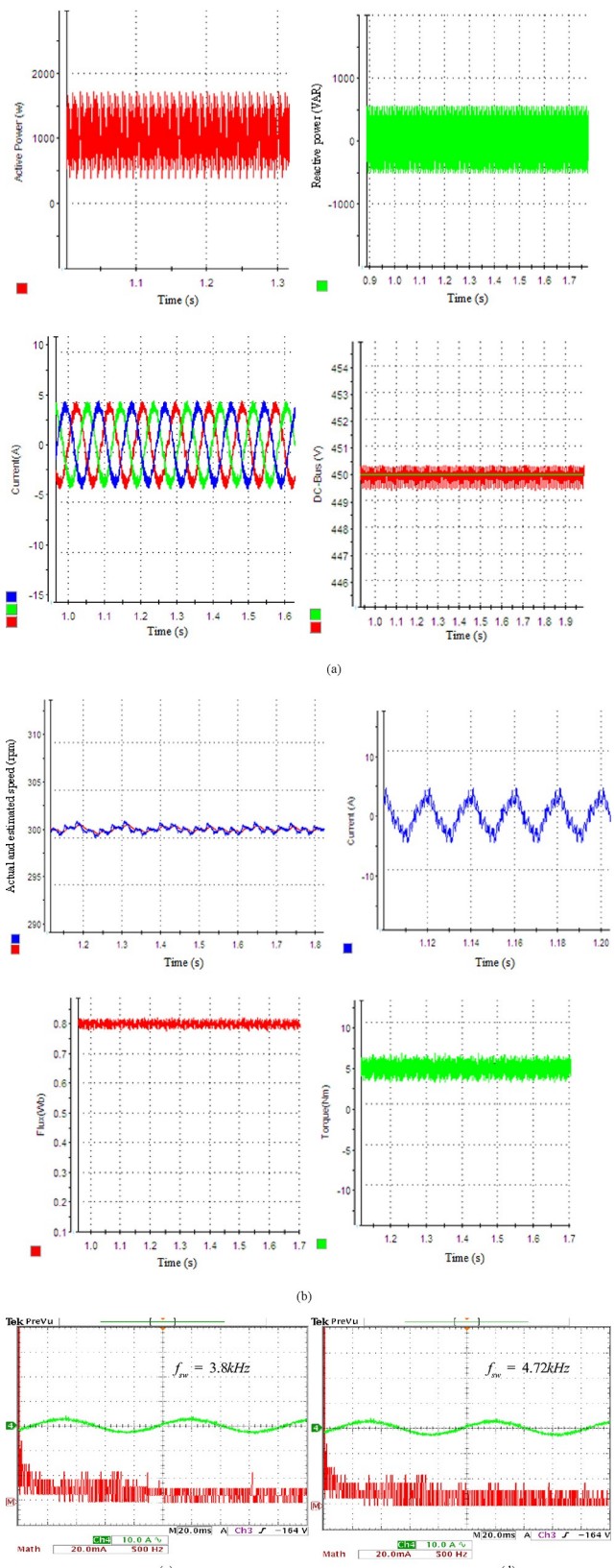

**Fig 6.** System's SSP: (a) rectifier stage by SSM-PPC technique, (b) MS by SPTC technique, (c) F-spectra of I's grid, (d) spectra of I's motor at 300 rpm.

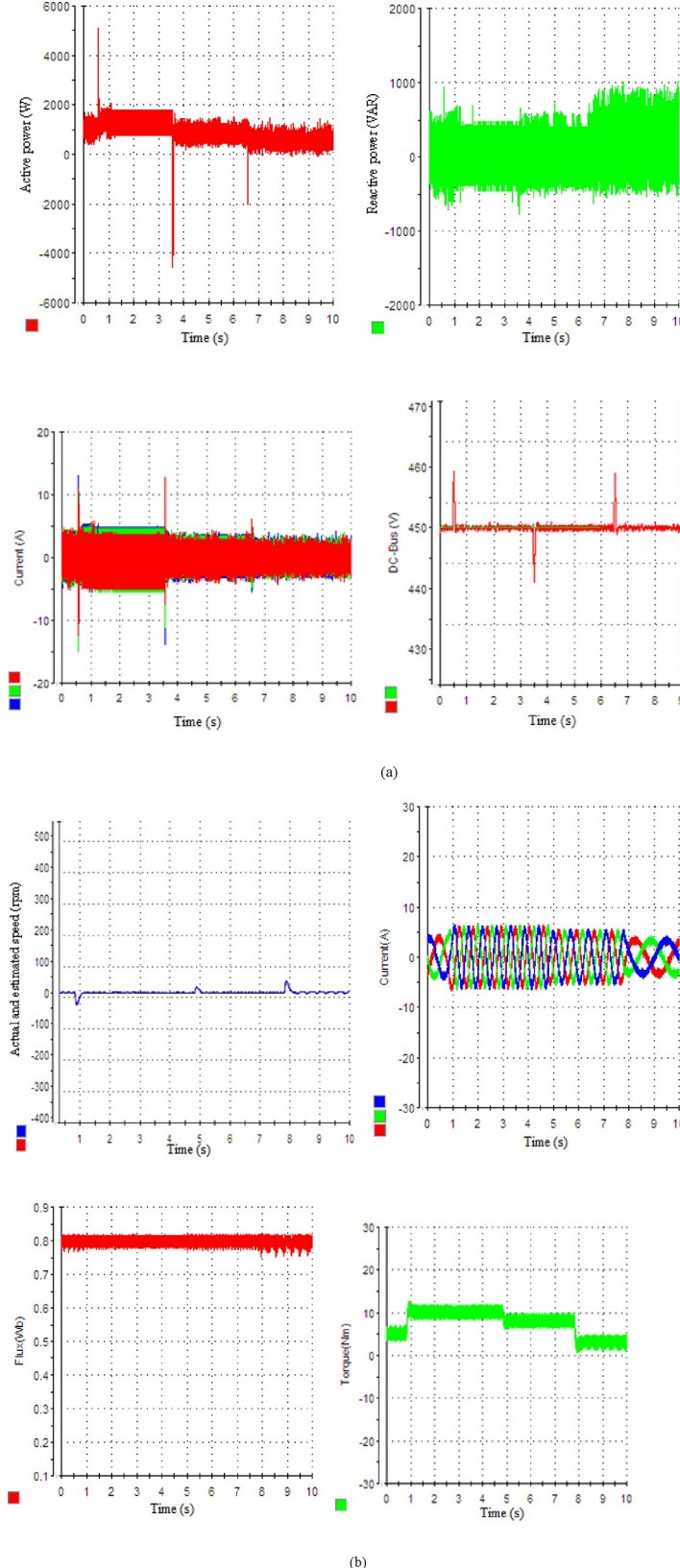

**Fig 7.** System's dynamic response: (a) rectifier side by SSM-PPC approach, (b) motor side by SPTC strategy.

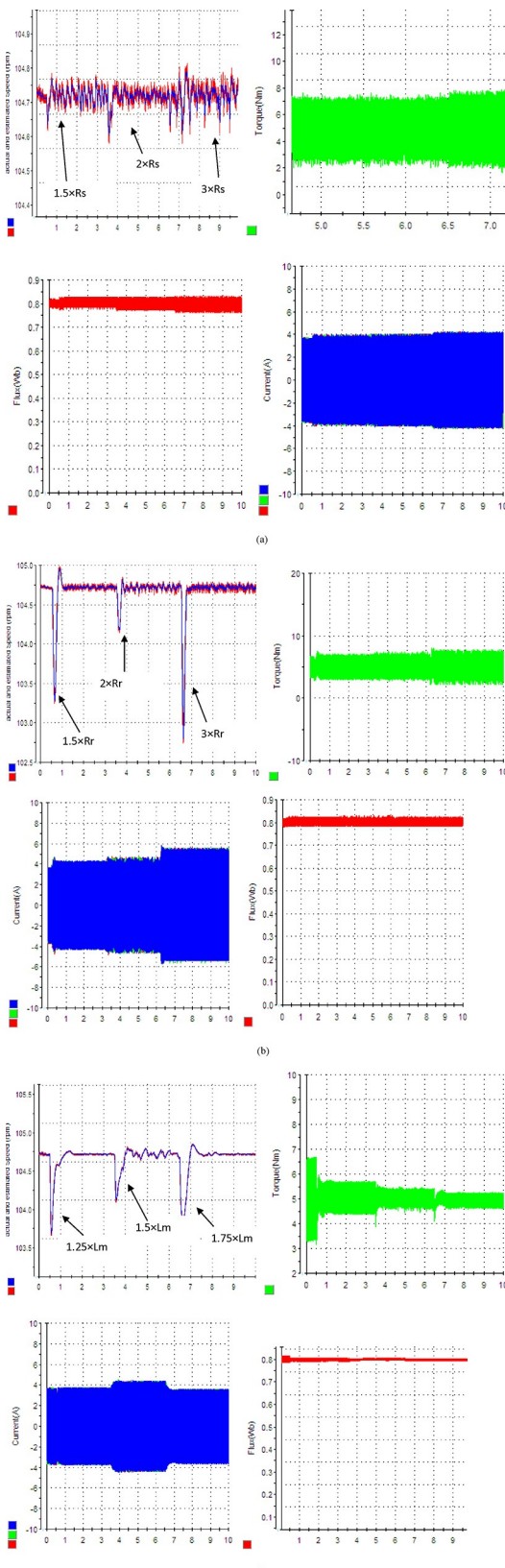

**Fig 8.** Estimation results under parameters variations (a) $R_s$ variation, (b) $R_r$ variation, and (c) $L_m$ variation.

**Table 6. Performance comparison with the recent works.**

| Refs. | Computational complexity | Inclusion of system nonlinearity | Dynamic response | WF | Parameter sensitivity | Switching frequency |
|---|---|---|---|---|---|---|
| [46] | High | Hard | Low | No | High | Variable |
| [47] | Low | Hard | Medium | No | High | Variable |
| [48] | High | Hard | Fast | Yes | High | Fixed |
| [49] | Low | Easy | Low | Yes | Low | Fixed |
| [50] | High | Hard | Medium | No | High | Variable |
| [51] | High | Easy | Low | No | Low | Variable |
| Current study | Low | Easy | Fast | No | Low | Fixed |

## Author Contributions

**Conceptualization:** Mohamed Chebaani, Noura A. Nouraldin.

**Formal analysis:** Ahmad F. Tazay.

**Funding acquisition:** Ahmad F. Tazay.

**Investigation:** Mohamed Metwally Mahmoud, Mohamed I. Mosaad.

**Methodology:** Mohamed Metwally Mahmoud, Noura A. Nouraldin.

**Resources:** Mohamed Metwally Mahmoud, Ahmad F. Tazay, Mohamed I. Mosaad, Noura A. Nouraldin.

**Software:** Mohamed Chebaani.

**Supervision:** Mohamed Metwally Mahmoud.

**Validation:** Ahmad F. Tazay.

**Visualization:** Mohamed Chebaani.

**Writing – original draft:** Mohamed Chebaani.

**Writing – review & editing:** Mohamed Metwally Mahmoud, Ahmad F. Tazay, Mohamed I. Mosaad.

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
