## [Decision Letter · Decision Letter 0]

1 Aug 2023

PONE-D-23-20791Extended Kalman Filter Design for Sensorless Sliding Mode Predictive Control of Induction Motors without Weighting Factor: An Experimental InvestigationPLOS ONE

Dear Dr. Mahmoud,

Thank you for submitting your manuscript to PLOS ONE. After careful consideration, we feel that it has merit but does not fully meet PLOS ONE’s publication criteria as it currently stands. Therefore, we invite you to submit a revised version of the manuscript that addresses the points raised during the review process.

We look forward to receiving your revised manuscript.

Kind regards,

Dhanamjayulu C, Ph.D & Post.Doc

Academic Editor

PLOS ONE

Journal Requirements:

Additional Editor Comments:

The reviewers recommend reconsideration of your manuscript following major revision and modification. I invite you to resubmit your manuscript after addressing the comments raised by the reviewers.

Reviewers' comments:

Reviewer's Responses to Questions

**Comments to the Author**

1. Is the manuscript technically sound, and do the data support the conclusions?

Reviewer #1: Yes

Reviewer #2: Yes

2. Has the statistical analysis been performed appropriately and rigorously? 

Reviewer #1: Yes

Reviewer #2: I Don't Know

3. Have the authors made all data underlying the findings in their manuscript fully available?

Reviewer #1: Yes

Reviewer #2: Yes

4. Is the manuscript presented in an intelligible fashion and written in standard English?

Reviewer #1: Yes

Reviewer #2: Yes

5. Review Comments to the Author

Reviewer #1: Dear Authors,

This manuscript presents an EKF design for sensorless sliding mode predictive control of IMs without the weighting factor. In this paper, the results are presented in a professional and scientific manner. However, the following comment should be taken into account.

• The performance of the proposed method against IM parameters (Rs Rr, and Lm) changes should be examined.

• If possible, a zero speed scenario should be added under different load torque variations.

• Results can be obtained in higher quality on Matlab/Simulink by using DS1104 instead of oscilloscope.

• The following recent studies from the studies with EKF can be given as a reference.

R. Demir, ‘Robust stator flux and load torque estimations for induction motor drives with EKF-based observer’, Electrical Engineering, vol. 105, no. 1, pp. 551–562, Feb. 2023, doi: 10.1007/s00202-022-01717-y.

E. Zerdali̇ and R. Demi̇r, ‘Speed-sensorless predictive torque controlled induction motor drive with feed-forward control of load torque for electric vehicle applications’, Turkish Journal of Electrical Engineering & Computer Sciences, vol. 29, no. 1, pp. 223–240, 2021, https://doi.org/10.3906/elk-2005-75

The study is valuable and the results are potentially useful. The article can be re-evaluated after revision in line with the comments.

Best regards.

Reviewer #2: I appreciate the hard work done by the authors. Having said that, the following comments bears significance, concerning this paper:

• Summarize the recent works in the form of a table.

• The results (curves and graphs) should be compared with the literature.

• What are the limitations of existing works that motivated the current research?

• What is the computational complexity of the proposed work?

• As a general note, the authors should consider using consistent and standardized mathematical symbols and annotations, e.g. using [.] for discrete-time representations and (.) for continuous-time representations.

• Why Extended Kalman Filter observer instead of the existing robust sliding mode observers? The motivation of using this observer is needed.

6. PLOS authors have the option to publish the peer review history of their article (what does this mean?). If published, this will include your full peer review and any attached files.

Reviewer #1: No

Reviewer #2: No

---

## [Author Response · Author response to Decision Letter 0]

26 Aug 2023

***Technical response to the reviewers*** August 27th, 2023

Journal: PLOS ONE

Manuscript No.: PONE-D-23-20791

Title: “Extended Kalman Filter Design for Sensorless Sliding Mode Predictive Control of Induction Motors without Weighting Factor: An Experimental Investigation” 

Mohamed Chebaani1, Mohamed Metwally Mahmoud2*, Ahmad F. Tazay3, Mohamed I. Mosaad 4,5

Noura A. Nour Aldin6

1 Department of Electrical Motorering, LGEB laboratory, Biskra University, Algeria

2Electrical Engineering Department, Faculty of Energy Engineering, Aswan University, Aswan 81528, Egypt

3 Electrical Engineering Department, Colleague of Engineering, Al Baha University, KSA

4 Electrical & Electronics Engineering Technology Department, Royal Commission Yanbu Colleges & Institutes,

Yanbu Industrial City 46452, Saudi Arabia

5 Electrical Engineering Department, Faculty of Engineering, Damietta University, Damietta 34511, Egypt

6 Electrical Department, Faculty of Technology and Education, Suez University, Suez 43533, Egypt

dr.chebanimohamed@gmail.com, metwally_m@aswu.edu.eg, afareed@bu.edu.sa, m_i_mosaad@hotmail.com, Noura.nouraldin@ind.suezuni.edu.eg

Corresponding authors: Mohamed Metwally Mahmoud (metwally_m@aswu.edu.eg), and Noura A. Nour Aldin (Noura.nouraldin@ind.suezuni.edu.eg)

Dear Editors and Reviewers

The authors are thankful to the learned Editor and Reviewers for their thoughtful and detailed comments to improve the quality of the manuscript. The authors have given reviewer comments a lot of interest in the revision process in an attempt to address all of the reviewers’ concerns and corrections as you will already find them incorporated in the revised manuscript. Moreover, a reply to each of the reviewers’ comments is provided below.

Kindly find the response to the reviewer’s comments in the following paragraphs. We hope this revised version of the manuscript meets the editor and reviewers’ expectations, and the standards of publication in the PLOS ONE Journal. 

The changes carried out by the authors are incorporated in the revised manuscript and highlighted in YELLOW.

Editor's Comments:

Comments to the Authors:

Comment-1: Thank you for submitting your manuscript to PLOS ONE. After careful consideration, we feel that it has merit but does not fully meet PLOS ONE’s publication criteria as it currently stands. Therefore, we invite you to submit a revised version of the manuscript that addresses the points raised during the review process. The paper should be revised thoroughly incorporating the comments offered by reviewers.

Response-1: Our sincere thanks and appreciation to the editor for considering our manuscript for publication in PLOS ONE Journal, and the recommending submission of the revised manuscript with major revisions. To improve the quality of the manuscript, the reviewer's queries are addressed and their suggestions are incorporated into the revised manuscript. The changes carried out by the authors are incorporated in the revised manuscript and highlighted in YELLOW to be easily viewed by the editors and reviewers. A cover letter is provided and prepared to explain, point by point, the details of the revisions to the manuscript. Kindly, check the revised version.

Reviewers Comments:

Reviewer#1 

Comments to the Authors:

Comment-1: The performance of the proposed method against IM parameters (Rs Rr, and Lm) 

 changes should be examined.

Response-1: First, the authors would like to thank the reviewers for their insightful comments and useful feedback that will enhance the presentation and quality of the paper. From our side, we are very keen to answer and take into account the reviewer's comments. Based on the reviewers' comments, the whole paper is revised with more clarifications and refinements. For example, the performance of the proposed method against IM parameters (Rs Rr, and Lm) changes are added. A new paragraph is added in the revised manuscript. Kindly refer to the

highlight text in Section 6.4. 

Comment-2: If possible, a zero-speed scenario should be added under different load torque variations.

Response-2: As recommended by the reviewer, a zero-speed scenario is added under different load torque variations. kindly refer to the highlighted text in the revised version, Section 6.3.

Comment-3: Results can be obtained in higher quality on MATLAB/Simulink by using DS1104 instead of oscilloscope.

Response-3: The authors thank the reviewer for his recommendation, which would enhance the quality of the paper. The results were implemented by using DS1104. We updated the manuscript. Kindly refer to the revised version, Section 6.1, 6.2, 6.3, and 6.4.

Comment-4: The following recent studies from the studies with EKF can be given as a reference.

Response-4: The authors thank the reviewer for his recommendation, which would enhance the quality of the paper. The article's references are updated and added the references suggested by reviewer. Kindly refer to the highlighted lines in the reference section.

1. R. Demir, “Robust stator flux and load torque estimations for induction motor drives with EKF-based observer,” Electr. Eng., vol. 105, no. 1, pp. 551–562, 2023, doi: 10.1007/s00202-022-01717-y.

2. E. Zerdali and R. Demir, “Speed-sensorless predictive torque controlled induction motor drive with feed-forward control of load torque for electric vehicle applications,” Turkish J. Electr. Eng. Comput. Sci., vol. 29, no. 1, pp. 223–240, 2021, doi: 10.3906/ELK-2005-75.

Reviewer#2 

Comments to the Authors:

Comment-1: Summarize the recent works in the form of a table.

Response-1: The reviewer here raises a very good point about the recent works in the form of a table and has been added. A new table is added about this point in the revised paper. Kindly refer

to the highlighted table.

Comment-2: The results (curves and graphs) should be compared with the literature.

Response-2: The authors express their gratitude to the reviewer for providing a valuable comment that will enhance the content of the paper. Obtaining the graphical data for alternative methodologies poses a challenging problem, particularly in the context of the experimental work being presented. Comparative research was conducted, wherein six additional techniques were included in Table 6 to demonstrate the efficacy of the suggested method.

Comment-3: What are the limitations of existing works that motivated the current research?

Response-3: The authors thank the reviewer for pointing this point out. Computational burden and designing a cost function are the implementation challenges of the predictive torque control (PTC) algorithm. The computational burden depends on the number of prediction vectors and the complexity of calculations mainly stator current and torque in the iterative prediction loop. Research has also been conducted on selecting the weighting factors used in the cost function. Because different control objectives in the cost function have different magnitudes and units. Hence, combining the control objectives in the cost function is a challenging task. In this paper, a stator flux-based equivalent reference stator flux vector calculation is proposed, to avoid the complex torque calculations in the prediction loop and thus the weighting factor tuning between torque and flux errors. Along with this RSFVC, selected prediction vectors (SPVs) are proposed for the prediction loop. This strategy reduces the number of iterations of the prediction loop; thus, the complexity of the prediction loop is reduced further. kindly check the updated manuscript. 

Comment-4: What is the computational complexity of the proposed work?

Response-4: The reviewer here raises a very good point about the computational complexity of the proposed work. The proposed control algorithm is implemented using a dSPACE DS1104 R&D controller board with Control Desk and MATLAB Simulink software packages. A time base counter is used for measurement of the execution time of the algorithms. Functions, like starting and reading the counter, are used to get the duration of a particular step of which the execution time is to be measured. The built-in PowerPC time base has a high resolution of 4/bus clock (40 ns). Finally, the measured execution is observed in Control Desk. 

The proposed algorithm requires additional calculations for multi-objective ranking method and. Extended Kalman Filter (EKF) observer. However, the calculations are very simple. To reduce the computational complexity and improve the dynamic estimation performance, a EKFs-based speed estimation method was realized, with one two-dimensional EKF operating in serial, so as to overcome the hysteretic nature of the traditional EKF. Therefore, the required maximum execution time is 115.6 μs. Most of the execution time is spent on the estimations (Torque and flux) and predictions: 27%and 62%, respectively. For this reason, it is possible to execute the proposed control algorithm within 130 μs. A new clarifications are added about this point in the revised paper. Kindly refer to the highlighted text, section 5.

Comment-5: As a general note, the authors should consider using consistent and standardized mathematical symbols and annotations, e.g. using [.] for discrete-time representations and (.) for continuous-time representations

Response-5: We are thankful for your concern on the manuscript. The revised manuscript is uploaded as per your instructions. kindly check the updated manuscript. 

Comment-6: Why Extended Kalman Filter observer instead of the existing robust sliding mode observers? The motivation of using this observer is needed.

Response-6: The authors thank the reviewers for pointing this point out. Actually, The Extended Kalman Filter formulated in this research is more accurate in terms of steady state speed estimation error, and usually has less standard deviation in such estimates as well, indicating a less oscillatory response. The Extended Kalman Filter is more effective at processing measurement noise than the Sliding Mode Observer. Between the choice of the Extended Kalman Filter and sliding Mode Observer, it is the author’s opinion that the Extended Kalman filter would be a more prudent choice for state estimation for the induction motor, assuming that its integration can be calculated taking into account the processing power of the dSPACE card. New clarifications are added about this point in the revised paper. Kindly refer to the highlighted text, section 4.

The authors once again thank the learned Editors and Reviewers for their valuable comments for improving the quality of the manuscript.

---

## [Decision Letter · Decision Letter 1]

10 Oct 2023

Extended Kalman Filter Design for Sensorless Sliding Mode Predictive Control of Induction Motors without Weighting Factor: An Experimental Investigation

PONE-D-23-20791R1

Dear Dr.

We’re pleased to inform you that your manuscript has been judged scientifically suitable for publication and will be formally accepted for publication once it meets all outstanding technical requirements.

Kind regards,

Dhanamjayulu C, Ph.D & Post.Doc

Academic Editor

PLOS ONE

Additional Editor Comments (optional):

The reviewers recommend reconsideration of the article for the publication.

The article can be accepted in the form.

Reviewers' comments:

Reviewer's Responses to Questions

**Comments to the Author**

1. If the authors have adequately addressed your comments raised in a previous round of review and you feel that this manuscript is now acceptable for publication, you may indicate that here to bypass the “Comments to the Author” section, enter your conflict of interest statement in the “Confidential to Editor” section, and submit your "Accept" recommendation.

Reviewer #1: All comments have been addressed

Reviewer #2: (No Response)

2. Is the manuscript technically sound, and do the data support the conclusions?

Reviewer #1: Yes

Reviewer #2: (No Response)

3. Has the statistical analysis been performed appropriately and rigorously? 

Reviewer #1: Yes

Reviewer #2: (No Response)

4. Have the authors made all data underlying the findings in their manuscript fully available?

Reviewer #1: Yes

Reviewer #2: (No Response)

5. Is the manuscript presented in an intelligible fashion and written in standard English?

Reviewer #1: Yes

Reviewer #2: (No Response)

6. Review Comments to the Author

Reviewer #1: (No Response)

Reviewer #2: The paper has undergone a thorough revision, and all concerns I raised in my previous review have been addressed.

7. PLOS authors have the option to publish the peer review history of their article (what does this mean?). If published, this will include your full peer review and any attached files.

Reviewer #1: No

Reviewer #2: No

---

## [Editor Report · Acceptance letter]

14 Nov 2023

PONE-D-23-20791R1 

Extended Kalman Filter Design for Sensorless Sliding Mode Predictive Control of Induction Motors without Weighting Factor: An Experimental Investigation 

Dear Dr. Mahmoud:

I'm pleased to inform you that your manuscript has been deemed suitable for publication in PLOS ONE. Congratulations! Your manuscript is now with our production department. 

Kind regards, 

on behalf of

Dr. Dhanamjayulu C 

Academic Editor

PLOS ONE